# Impact of Changed Use of Greenspace during COVID-19 Pandemic on Depression and Anxiety

**DOI:** 10.3390/ijerph18115842

**Published:** 2021-05-29

**Authors:** Seulkee Heo, Miraj U. Desai, Sarah R. Lowe, Michelle L. Bell

**Affiliations:** 1School of the Environment, Yale University, New Haven, CT 06511, USA; michelle.bell@yale.edu; 2Department of Psychiatry, School of Medicine, Yale University, New Haven, CT 06511, USA; miraj.desai@yale.edu; 3Department of Social and Behavioral Sciences, School of Public Health, Yale University, New Haven, CT 06510, USA; sarah.lowe@yale.edu

**Keywords:** greenspace, mental health, Coronavirus, SARS-CoV-2, social determinants of mental health, intervention, pandemic, South Korea

## Abstract

The COVID-19 pandemic has had devastating consequences for health, social, and economic domains, but what has received far less focus is the effect on people’s relationship to vital ecological supports, including access to greenspace. We assessed patterns of greenspace use in relation to individual and environmental factors and their relationship with experiencing psychological symptoms under the pandemic. We conducted an online survey recruiting participants from social media for adults in Korea for September–December 2020. The survey collected data on demographics, patterns of using greenspace during the pandemic, and major depression (MD) and generalized anxiety disorder (GAD) symptoms. The Patient Health Questionnaire (PHQ-9) and the Generalized Anxiety Disorder 2-item (GAD-2) were applied to identify probable cases of MD and GAD. A logistic regression model assessed the association decreased visits to greenspace after the outbreak compared to 2019 and probable MD and GAD. Among the 322 survey participants, prevalence of probable MD and GAD were 19.3% and 14.9%, respectively. High rates of probable MD (23.3%) and GAD (19.4%) were found among persons currently having job-related and financial issues. Of the total participants, 64.9% reported decreased visits to greenspace after the COVID-19 outbreak. Persons with decreased visits to greenspace had 2.06 higher odds (95% CI: 0.91, 4.67, significant at *p* < 0.10) of probable MD at the time of the survey than persons whose visits to greenspace increased or did not change. Decreased visits to greenspace were not significantly associated with GAD (OR = 1.45, 95% CI: 0.63, 3.34). Findings suggest that barriers to greenspace use could deprive people of mental health benefits and affect mental health during pandemic; an alternative explanation is that those experiencing poor mental health may be less likely to visit greenspaces during pandemic. This implies the need of adequate interventions on greenspace uses under an outbreak especially focusing on how low-income populations may be more adversely affected by a pandemic and its policy responses.

## 1. Introduction

A growing body of literature suggests that greenspace can promote human health through psychological and physical health benefits [1]. In particular, a major health benefit of greenspace is mental health, with documented links to lower risk of major depression (MD) and generalized anxiety disorder (GAD) [2]. Greenspace can be defined as natural vegetation such as grass, plants, or trees and built green structures such as parks. Potential pathways for mental health effects of greenspace are restorative effects from stress relief and mood enhancement [3], physical activities, neighborhood social cohesion (e.g., social contact) [4,5], and reduced exposure to environmental hazards such as air pollution and noise [5]. Physical activity may mediate the relationship between exposure to greenspace and mental health. Viewing nature can trigger positive emotions and relieve stress (i.e., stress reduction theory) [6]. Furthermore, nature can hold a person’s effortless attention and thereby help reduce mental fatigue and stress [7]. Facilitated neighborhood social cohesion in greenspace through socializing and holding public events is a potential mediator of the mental health benefits of greenspace [4,6]. People living in communities with high social cohesion can enjoy social support and are more exposed to health-enhancing activities [8]. Evidence of health effects of greenspace is of high interest to urban planners and decision-makers seeking nature-based solutions for health challenges in cities [9], alongside a broader response to the climate and ecological crisis. Therefore, understanding the role of greenspace on human health, especially on mental health, is important to aid urban planning of greens and open space under rapid urbanization.

### COVID-19 and Mental Health

The novel coronavirus (SARS-CoV-2) outbreak was identified in January 2020 in China [10] and The World Health Organization (WHO) announced the COVID-19 pandemic on March 11, 2020 [11] Pandemic mitigation measures such as physical/social distancing interventions, while intending to reduce virus transmission may be detrimental to mental health and well-being, which are already hampered under public health and economic crisis conditions. A few articles suggested that the prolonged stay-at-home orders could lead to adverse mental health outcomes such as MD, GAD, and loneliness [12,13,14,15]. Increased prevalence of MD and GAD after the outbreak was reported in some countries including Canada, China, and Greece [16,17,18,19]. Some epidemiologic studies assessing the impacts of COVID-19 outbreak on mental health outcomes [20,21,22,23,24,25,26,27] suggested that female gender, unemployment, poor self-reported health status, loneliness, poor social supports, and contact with suspected infection or COVID-19 confirmed cases were associated with greater risks of psychological distress from the pandemic. Identifying high-risk groups of mental health outcomes is imperative for developing psychological interventions. However, evidence of the interrelationships between COVID-19, mental health, and greenspace—earlier identified as beneficial for well-being—is limited.

A less explored factor that could influence mental health during the COVID-19 pandemic is the use of greenspace. The COVID-19 pandemic has resulted in a new way of looking at the social and public health values of greenspace for urban dwellers [28]. There are at least four potential pathways through which our interactions with greenspace could be altered by COVID-19 and thereby affect human health under the pandemic (Figure 1). First, the use of greenspace may decrease due to mitigation measures during the pandemic and safety concerns, thereby lowering the associated mental health benefits [28]. Governments and communities have applied physical/social distancing in different ways including closing urban greenspace (e.g., parks) and limiting visitors, and these measures have constantly changed in accordance with the severity of disease spread. Second, fear of contracting COVID-19 in greenspace may discourage visits. Third, those who suffer from financial, employment, and mental health challenges may be especially less likely to visit greenspace during pandemic. For example, persons with MD might be less likely to initiate pleasurable or restorative behaviors and persons with GAD might be likely to avoid greenspace due to fears for COVID-19. The potential loss or change of visits to greenspace through these three pathways may deprive people of potential mental health benefits, increasing risk for MD and GAD symptoms.

On the other hand, for some persons, particularly those with adequate access and resources, the use of greenspace may increase during the pandemic as other activities are limited or prohibited. Due to several alterations of daily lifestyles such as the transition to working from home, some people may be more likely to visit outdoor public locations including parks that were open during the pandemic [29]. Increased Internet search keywords such as “go for a walk” during the pandemic indicates the desire of people to go outside for part of the day while they are spending more time at home [28]. Those who can visit greenspace during the pandemic may gain protective mental health effects compared to those who do not or cannot. Although evidence is extremely limited, a few reports have suggested that since COVID-19 control measures, mobility to parks increased in some European cities and in the US. A study indicated that in the US, while mobility generally decreased during the pandemic compared to earlier (e.g., 13% lower for retail and recreation), mobility to parks increased 54% [30]. A recent US study suggested that daily human mobility during the early phase of the COVID-19 outbreak (March–April 2020) decreased less in communities with a higher amount of vegetation after stay-at-home orders indicating the potential impact of and disparities in greenspace access during a pandemic [31]. To date, almost no research has studied the intersection of these complex systems to understand how the pandemic affected the use of greenspace and the associated psychosocial effects.

When examining potential health impacts, it is important to consider a number of dimensions of greenspace usage, such as frequency, quality, and type [3]. Generally, research in this area has focused on measuring the amount of surrounding greenspace or spatial accessibility to greenspace [3]. However, the presence of greenspace, such as urban parks, may not reflect the frequency, duration, and purposes of visiting greenspace, which are associated with the mental health benefits of greenspace. For example, a study in England demonstrated that the least frequent users of greenspace showed the lowest psychological well-being [32], indicating that greenspace is beneficial for mental health, that those with better psychological well-being are more likely to frequently visit greenspace, or both. A study of four European cities found significant associations between better mental health status (lower nervousness and depression) and time spent in greenspace [33]. A recent global survey study found increased visits to nature and mental health benefits from visiting nature during the pandemic in some western countries [34]. Despite these findings, much is unknown regarding the mental health implications through interactions (i.e., contact) between greenspace and people, and such analyses are hindered by a lack of data. Research is needed that goes beyond the measures of amount of greenspace and considers more information on patterns, characteristics, and disparities of visiting greenspace. Further, evidence is needed on whether people visit greenspace more or less under the pandemic, how those visits differ in terms of purpose of visit and type of greenspace, and how these changes to visits of greenspace differentially impact mental health and well-being.

Access to greenspace appears to be unequal among population groups, with more greenspace in richer areas [35]. An increased amount of local greenspace is associated with more visits to greenspace and physical activities [36,37,38,39]. Thus, an important question is how socioeconomic and demographic characteristics of individuals are related to how often people use greenspace and the types of visits. While studies of environmental justice suggested that persons with lower socioeconomic status (SES) or living in deprived regions have less access to but higher dependence on public greenspace [40,41,42,43], it is unclear how the use of greenspace by different communities changes under a pandemic.

In this study, we explored the associations between changed greenspace use and mental health outcomes during the COVID-19 pandemic in South Korea and how such associations differed by individual-level characteristics and local amount of greenspace. In assessing potential changes in patterns of using greenspace, we focused on changes in visits to greenspace during the COVID-19 pandemic compared to the year 2019 (i.e., the pre-pandemic period). We also investigated the main factors that affected change in visits to greenspace during the COVID-19 outbreak compared to pre-pandemic periods. This study can aid decision-makers regarding guidelines for disease prevention and control, support for mental health care, urban forest and park management, and urban planning.

## 2. Materials and Methods

### 2.1. Survey Data

This study was approved by the institutional review board of Yale University (IRB# 2000028853 approved on 21 August 2020). Data were collected through an online anonymous survey that was conducted for adults age ≥19 living in South Korea recruited from social media platforms between 21 September and 7 December 2020. The online questionnaire was distributed through social media advertisement campaigns of Facebook and Instagram. For Facebook campaigns, several keywords were used to better target Facebook users who would likely click the survey link: Extreme sport, garden, health promoting, mountain, natural environment, outdoors, park, survey, travel, leisure, fitness, environmentalism, environmental movement, camping, outdoor recreation. Participants completed a structured questionnaire in Korean and agreed to the study through an online informed consent form.

The online questionnaire consisted of 4 domains: General use of greenspace focusing on the year 2019 (pre-pandemic period), effects of COVID-19 on greenspace use focusing on the year 2020 after the Coronavirus outbreak, mental health, and basic information (demographic and socioeconomic data). In the health domain, the questionnaire consisted of items for experiences of psychological symptoms based on the standardized screening tools as described below. The questionnaire collected data for sex, age (5-year intervals between 19 and 74, 75 or more), height, body weight, the highest level of completed education (none, elementary school, middle school, high school, college, bachelor’s degree, graduate school), smoking status, alcohol use, annual income (6–12 million, 12–24 million, 24–36 million, 36–48 million, 48–60 million, 60–72 million, 72–96 million, 96–120 million, and 120 million or more Korean Won), marital status (single, married, divorced, separated, widowed), ZIP-code of home address, and whether or not they were living at the same address in the year 2019 and 2020. Education and income served as a proxy for socioeconomic status (SES).

Of the 700 participants surveyed, 382 provided the ZIP-code where they live, which was essential for estimating ZIP-code level greenness. The final sample consisted of 322 participants who lived at the same address in 2019 and in 2020 at the time of survey. Based on the ZIP-code, participants who live in the 7 metropolitan cities (Seoul, Incheon, Daejeon, Daegu, Busan, Gwangju, Ulsan) were defined as urban dwellers and the other participants were defined as dwellers in rural areas.

### 2.2. Assessment of Mental Health

The experience of probable MD and GAD was assessed using the Patient Health Questionnaire (PHQ-9) and the Generalized Anxiety Disorder 2-item (GAD-2), respectively. Survey participants reported whether they experienced each symptom in the past 2 weeks and those who answered ‘yes’ were asked to rate the extent to which they experienced among “several days”, “more than half the days”, and “nearly every day” within the past 2 weeks. For MD, we summed the responses from the 9 items of the PHQ-9 to create a score ranging from 0–27. Those who had a score of 10 or higher were defined as having probable MD. For GAD, we also summed the score from the 2 items of the GAD-2 and a cutoff score of 3 was used to determine probable cases of GAD. Previous work showed that a threshold of 10 of the PHQ-9 has a sensitivity of 0.88 and a specificity of 0.89 for MD [44]. The PHQ-9 has been verified to be reliable and valid to screen probable MD patients in Korean population [45,46]. The questionnaire collected data on history of depression and anxiety based on the question asking ‘during last year (2019), have you experienced depression lasting more than 2 weeks that affected your daily life? (yes, no, not sure)’.

As other factors related to the COVID-19 pandemic or attributable to the COVID-19 pandemic could affect mental health, we collected data from an open-ended optional question asking ‘what are the things you are most concerned about in your life right now?’. Using the collected answers, we identified 5 major types of concerns: Concerns about life or future, health-related concerns (including general health and COVID-19), job-related or financial concerns, family-related concerns (regarding health, future, well-being), and concerns for environment and society. We created categorical binary variables for these 5 types of concerns (1 for ‘yes’ and 0 for ‘no’) for having a given type of concern. We further categorized the concern type of ‘job-related or financial concerns’ into sub-types with ‘job search’, ‘job (currently employed)’, ‘general financial issues’, and ‘financial difficulties due to COVID-19′.

### 2.3. Assessment of Use of Greenspace and Local Greenspace

We collected data for various aspects of greenspace use from the survey participants. The questionnaire asked questions including “in general, which type of greenspace did you visit in the year 2019”, “how many times did you visit greenspace”, “how long did you spend in greenspace on average per visit”, “what were your main reasons to visit greenspace”, “in general, with whom do you visit greenspace”, “how safe did you feel in your neighborhood greenspace in the year 2019”, and “how much do you care about being able to visit greenspace as a part of your lifestyle?” For the question on the purposes of visiting greenspace, we asked the same question separately for the year 2019 and 2020 after the COVID-19 outbreak. Participants were allowed to choose multiple responses for this question among ‘relaxation’, ‘viewing nature’, ‘stress relief’, ‘leisure’, ‘exercise (e.g., walking, jogging)’, ‘walking pets’, ‘spending time with friends or family’, ‘public events’, and ‘other’. We assessed changes in visiting greenspace among survey participants by asking “have your visits to green space changed in 2020 since the Coronavirus outbreak compared to 2019?” with answers among ‘significantly increased’, ‘slightly increased’, ‘unchanged’, ‘slightly decreased’, and ‘significantly decreased’. Participants who chose the answers of ‘slightly decreased’ and ‘significantly decreased’ were defined as a group with perceived decreases in frequency of visiting greenspace. Participants with the other answers were defined as a group with unchanged or increased frequency of visiting greenspace.

To explore factors associated with the changes in frequency of visiting greenspace, participants reporting decreased visits to greenspace after the Coronavirus outbreak were asked the follow-up question of “if your visits to green space changed in 2020 compared to 2019, what factors changed your decision to go to greenspace in 2020?” Participants were permitted to select multiple choices among ‘fear and anxiety about Coronavirus’, ‘government urged to stay at home’, ‘closure of greenspace due to Coronavirus’, ‘just don’t feel like visiting’, ‘increased crowding in greenspace’, ‘issues unrelated to the Coronavirus’, and ‘disease, disorder, or injury (not related to COVID-19)’.

We estimated greenspace for each ZIP code using a vegetation index using the 250-m resolution Enhanced Vegetation Index (EVI) 16-day composite data from the Moderate Resolution Imaging Spectroradiometer product MOD13Q1 [47]. We estimated ZIP-code vegetation level by calculating the average of the pixel values of the EVI data within and surrounding the area’s boundary for each ZIP code area. Reprojection and mosaicking were conducted at the Google Engine program and the calculation of EVI was conducted using the R statistical program.

### 2.4. Statistical Analysis

Statistical analyses were applied to assess the following: (1) Factors are associated with changes in visits to greenspace, (2) associations between the changes in visits to greenspace and risks of probable MD and GAD, and (3) effect modifiers for such associations.

To assess the factors associated with changes in visits to greenspace, we performed the chi-square test. This was applied by age, gender, education, income, marital status, urbanicity, perceived safety in neighborhood greenspace, importance of using greenspace in life, purposes of visiting greenspace, and type of greenspace visited. Descriptive summarization was conducted for self-reported reasons for changes in visits to greenspace for the subset of participants with decreased visits to greenspace.

We applied a multivariable binary logistic regression analysis to assess the associations between the changes in frequency of visiting greenspace (unchanged or increased visits vs. decreased visits), comparing the pandemic period to the pre-pandemic period, and probable MD and GAD. Odds Ratios (ORs) and 95% Confidence Intervals (CIs) are presented. The following characteristics were controlled for as potential confounders: Age (19–29, 30–49, 50–64, 65+ years), gender (men, women, other), smoking (former smoker, current smoker, never smoker), marital status (single, widowed/divorced/separated), experience of depression in the year 2019 (yes, no), experience of anxiety in the past 12 months (yes, no), frequency of visiting greenspace in the year 2019 (0–2 times per month, 1–2 times per week, 3–4 times per week, 5 times or more per week), having job-related or financial concerns (yes, no), having health-related concerns (yes, no), and body mass index (BMI). Alcohol use was not considered in the model due to its high correlation with smoking (chi-squared test, *p*-value = 0.008). Alcohol use was excluded after conducting a stepwise regression analysis. Frequency of visiting greenspace in last year (2019) was adjusted as a confounder. The rationales of this confounder are that increases in frequency of visiting/using greenspace is significantly associated with better mental health [33] and is potentially correlated with likelihood of using greenspace. Frequent users of greenspace (e.g., parks) in the pre-pandemic period would be less likely to refrain from visiting greenspace during pandemic than less frequent users. The correlation between the changes in visits to greenspace and frequency of visiting greenspace in 2019 (i.e., pre-pandemic period) was (*p*-value = 0.015). For probable GAD, education level was excluded in the logistic regression model since over-fitting of the model (i.e., including more parameters than can be justified by the sample data) occurred.

In the sensitivity analysis, we estimated ORs of probable MD and GAD for decreased visits and increased visits to greenspace separately in comparison to unchanged visits to greenspace as the reference.

Effect modification of the impact of changed visits to greenspace on probable MD and GAD by type of greenspace visited and purposes of visiting greenspace was investigated by applying an interaction term between the variable of changes in visits to greenspace and potential effect modifiers in the logistic regression models. Categorization of purposes of visiting greenspace was conducted based on psychological purposes (e.g., relaxation, viewing nature, stress relief, leisure), physical activities (e.g., exercise, walking pets), and social interactions (e.g., spending time with friends or family, public events such as outdoor concert). We also assessed effect modification by education level, urbanicity, perceived safety level in neighborhood greenspace, and importance of using greenspace in life.

## 3. Results

The prevalence of probable MD was 19.3% (*n* = 62), and the prevalence of probable GAD was 14.9% (*n* = 48). Probable cases of MD were higher in women (*n* = 52, 16.2%) than men (*n* = 8, 2.5%). Demographic characteristics and other descriptive data for the study participants divided by probable cases of MD and GAD are presented in Table 1. Of the total sample (*n* = 322), 76 (23.8%) participants were male. The survey included a higher percentage of women than the local population (75.9% vs. 50.1% for mid-2019) [48]. Almost half of the participants were in the age group 30–49 years (*n* = 155, 48.1%). Of the total number of participants, 57.5% (*n* = 176) lived in urban areas, 72.7% (*n* = 224) had an undergraduate school degree or higher, and 52.2% (*n* = 163) were single. The mean of EVI across all participants was 0.17. Women, persons with middle-high school education, and single persons had significantly higher prevalence of probable MD and GAD.

We examined whether visits to greenspace decreased during the pandemic period compared to the pre-pandemic period, and how such changes to the frequency of greenspace visits varies by individual characteristics and purpose of visits. Across all study participants, 64.9% (*n* = 209) reported that their visits to greenspace decreased in 2020 under the pandemic compared to year 2019; 118 (36.7%) persons reported ‘slightly decreased’ and 91 (28.3%) persons reported ‘significantly decreased’. Of the participants, 15.5% (*n* = 50) reported unchanged frequency of visits to greenspace, 7.5% (*n* = 24) reported ‘significantly increased’ visits, and 12.1% (*n* = 39) reported ‘slightly increased’ visits. Persons aged 39 to 49 years compared to the other age groups showed significantly higher decreases in visits to greenspace under the pandemic compared to the pre-pandemic period (*p*-value = 0.001) (Appendix A). Persons who used greenspace for social interactions before the pandemic (i.e., year 2019) showed significantly higher decreases in visits to greenspace than persons who did not use greenspace for social interactions (*p*-value = 0.004) (Appendix A).

Among participants whose visits to greenspace decreased in 2020 compared to 2019 (*n* = 209), we examined which factors influenced their decreased tendency of visiting greenspace during the pandemic compared to the pre-pandemic period (Figure 2). The most predominant factor was ‘fear and anxiety about Coronavirus’ (*n* = 169, 80.9%) followed by ‘government urged to stay at home’ (*n* = 151, 72.2%). About 30% of participants reported that their visits to greenspace during the pandemic compared to 2019 decreased due to increased crowding in greenspace (*n* = 57) or closure of greenspace due to Coronavirus (*n* = 57). Suspended public transportation was not a major factor to reduce visits to greenspace among the survey participants.

We examined if the main purposes of visiting greenspace differed during the pandemic (2020) and pre-pandemic period (2019) as shown in Figure 3. During the pandemic, respondents were more likely to visit greenspace for stress relief (52.2% [*n* = 168] during the pandemic compared to 50.3% [*n* = 162] pre-pandemic) and ‘other’ reasons (17.1% [*n* = 55] during the pandemic compared to 13.0% [*n* = 42] pre-pandemic). During the pandemic compared to 2019, fewer respondents reported visiting greenspace for relaxation, viewing nature, spending time with friends or family, leisure, exercise, or public events.

Based on the open-ended question (“what are the things you are most concerned about in your life right now?”), we identified five categories of concerns: Concerns about life or future, health-related concerns (including general health and COVID-19), job-related or financial concerns, family-related concerns (regarding health, future, well-being), and concerns for environment and society. We examined the frequency of these categories. Among the identified five types of concerns at the time of survey, the concern ‘job-related or financial concerns’ showed the highest prevalence among the survey participants (Appendix A).

Within the open-ended question regarding the biggest concern in life, some survey participants noted more than one of the five identified types of concerns. We examined correlation among the five categories of concerns. Among the 103 persons reporting job- or finance-related concerns, 9 (8.7%) persons also reported life- or future-related concerns, while 7 (6.8%) persons also reported concerns for health or COVID-19. The job-related or finance-related concerns were statistically correlated with life- or future-related concerns based on chi-square test (*p*-value < 0.05). We assessed the prevalence of probable MD and GAD among the persons reporting each type of concern (Appendix A). Among the five concern types, job-related or financial concerns showed the highest prevalence of probable MD and GAD; 23.3% (*n* = 24) had probable MD and 19.4% (*n* = 20) had probable GAD.

We estimated the percentage of probable MD and GAD cases among the participants who reported the five identified categories of concerns. The job-related or financial concerns were further grouped into ‘financial difficulties due to COVID-19’, ‘general financial issues’, ‘job (currently employed)’, and ‘job search’. Results showed that 50% (*n* = 2) of the persons who reported financial difficulties due to COVID-19 had MD, and these 50% had anxiety as well (Appendix A). While 25.0% (*n* = 1) of participants reporting concerns in their current job had probable MD, the rate of probably anxiety cases in those participants was 50.0% (*n* = 2).

The association between visits to greenspace and risk of MD and GAD adjusted for other greenness-related factors are shown in Table 2. Individuals with decreased visits to greenspace during the pandemic compared to the previous year (i.e., pre-pandemic period) showed about 2 times higher odds for experiencing probable MD than those with unchanged or increased visits to greenspace (OR = 2.06, 95% CI: 0.91, 4.67, statistically significant at *p* < 0.10). Decreased visits to greenspace did not show a significant association with the odds of probable GAD (OR = 1.45, 95% CI: 0.63, 3.34). The effects of potential confounders in the same models are shown in Appendix A. Age, sex, smoking status, BMI, education level, marital status, experience of depression in last year, having health-related concerns, and having job-related or financial concerns were controlled in the models for probable MD. An interquartile range (IQR) increase in BMI (OR = 1.49, 95% CI: 1.01, 2.19 for 4.6 increase in BMI) and experiencing depression in last year (OR = 7.01, 95% CI: 3.20, 15.37) were significantly associated with probable MD and the other confounders were not statistically associated with probable MD. The same confounders except for education level were controlled in the models for probable GAD. An IQR (4.6) increase in BMI was significantly associated with experiencing probable GAD (OR = 1.90, 95% CI: 1.30, 2.77). In sensitivity analysis, we conducted the regression analysis comparing the risks of probable MD and GAD in three groups for greenspace use during the pandemic (i.e., unchanged, increased, decreased). When unchanged visits to greenspace was used as the reference group, OR of probable MD was significant for decreased visits to greenspace (OR = 5.00, 95% CI: 1.22, 20.5, significant at *p* < 0.05), whereas OR of probable MD was not significant for increased visits to greenspace (OR = 3.84, 95% CI: 0.81, 18.25). Neither decreased visits to greenspace (OR = 1.71, 95% CI: 0.48, 6.11) nor increased visits to greenspace (OR = 1.54, 95% CI: 0.36, 6.55) was significantly associated with probable GAD.

We did not find significant effect modification of decreased visits to greenspace during the pandemic compared to pre-pandemic periods by purposes of visiting greenspace, safety level in neighborhood greenspace, or urbanicity (Appendix A). The association between decreased visits to greenspace and probable MD was significant for those who used greenspace for social interactions (OR = 2.82, 95% CI: 1.06, 7.53) and for those who live in rural regions (OR = 6.89, 95% CI: 1.24, 38.31). The positive associations between probable MD and decreased visits to greenspace were not significant for those who use greenspace for physical or psychological purposes. Safety level in neighborhood greenspace did not modify the associations between probable MD and decreased visits to greenspace. In addition, there was no significant effect modification for decreased visits to greenspace and probable MD by type of visited greenspace and personal importance of using greenspace among the survey participants (data not shown).

## 4. Discussion

We studied changed use of greenspace during the COVID-19 pandemic in Korean population and found decreased visits to greenspace among the stud participants. It was found that the stay-at-home order, physical/social distancing interventions, and fear and anxiety for COVID-19 were the primary factors for the decreased use of greenspace during the pandemic as suggested by the potential mechanisms. Our results also indicate that study participants may have visited greenspace for stress relief after the disease outbreak compared to the previous year (i.e., year 2019) in spite of overall decreases in visits to greenspace. While the increase in visits to greenspace for ‘stress relief’ for the pandemic were slightly increased compared to the previous period, such an increase is notable given that visits for all other specific reasons for visiting greenspace decreased, with statistically significant decreases for some purposes. For example, decreases in using greenspace for ‘relaxation’ and ‘spending time with friends or family’ were statistically significant based on chi-square test (*p* < 0.001), whereas visits for ‘stress relief’ showed a slight increase. This may indicate that people may seek greenspace to deal with stress during the outbreaks in which their physical/social activities are profoundly restricted as a consequence of quarantine and the stay-at-home orders.

Many studies have suggested mental health benefits of greenspace although there is no ultimate consensus on those pathways [5], and multiple pathways (e.g., psychological, physical, and social pathways) may be relevant. These pathways may be diminished, resulting in lower health benefits, if visits to greenspace decrease under an unusual condition such as pandemic. Almost no previous study has investigated how pandemic conditions affect these pathways and mental health effects to date. In our study, we explored several purposes of visiting greenspace in the survey questionnaire and categorized them in to three major categories (i.e., psychological, physical, social). These different purposes of visiting greenspace modified the relationships between risks of probable MD and GAD and decreased visits to greenspace although the effect modification was not statistically significant. The risk of probable MD associated with decreased visits to greenspace was lower in the group using greenspace for psychological or physical purposes than the group not using greenspace for these reasons. A potential reason for this finding could be increased indoor physical activities (likely at home), which would mitigate general stress and psychological distress from COVID-19 [15]. The risk of probable MD was higher in the group using greenspace for social interactions than the group not using greenspace for such reason, which differs from findings of a previous study suggesting no significant association between greenspace and social support in pre-pandemic periods in Spain [43]. Our result indicates that persons who use greenspace as a place for social interactions or cohesion may be more deprived of the mental health benefits during the pandemic with mandatory physical/social distancing.

However, we highlight the challenges in distinguishing effect modifications by these pathways from each other due to multiple interrelationships among the mediators and mental health outcomes. For example, running in an urban park can simultaneously enhance physical health (e.g., improving cardiovascular fitness and muscles, maintaining a healthy weight), enhance mood, and reduce stress. These mediators may also correlate to each other [49]. Using greenspace for both physical and psychological purposes may be more enhanced during disease outbreaks if indoor places for physical activities and exercise are closed. Moreover, people in Asian cultures may not neatly separate physical from psychological, which is more standard within western societies [50]. Although our main focus was to understand the mental health effect of changed visits to greenspace during the pandemic rather than to identify significant mediating factors, further study could test for mediation more explicitly and aid an understanding of through which pathways the mental health benefits of greenspace can occur.

An additional effect modifier in the study is urbanicity. According to the literature, urban dwellers experience more serious psychological symptoms such as depression and anxiety than people living in rural areas [51]. Some research examined the difference of health benefits between urban and rural greenspace, while results are inconsistent [52]. A few studies suggested that urban greenspace (e.g., parks) provide fewer restorative benefits than more natural environment (e.g., agricultural land, forest), possibly due to the smaller size of greenspace and lower biodiversity [52,53,54]. In our study, we did not find different patterns of using greenspace in the pre-pandemic period (e.g., frequency and duration in 2019) between urban major cities and rural areas. However, the association between decreased visits to greenspace and probable MD was significant in rural regions while the association was not significant for urban regions. Reasons for this difference are unclear, but types of greenspace and duration of staying in greenspace between urban and rural regions may play a role in different mental health effects of greenspace. For example, people tend to travel further for rural or country-side greenspace and spend longer time viewing nature [52]. While we focused on MD and GAD, the use of various measures of mental well-being such as life satisfaction may lead to different health effects of urban and rural greenspace as well. Further research is needed to understand the health effects of urban and rural greenspace including differences by type and use of greenspace.

Studies have investigated how SES and demographic factors alter the frequency of visiting greenspace of individuals and results are inconclusive [55,56]. We assessed the changes in frequency of visits to greenspace during the pandemic in relation to SES and demographic information of the survey participants. Our data showed that the rate of probable MD cases among those who reported their concerns on either general or current finances (24.3%) was higher than the rate for the total survey participants (19.3%). Results showed that decreases in visits to greenspace significantly differed by age and income level. People age 30 to 49 years had higher reduction in visits to greenspace during the pandemic compared to other age groups. People with annual income between 24 and 74 million KRW reported higher rates of decreased visits to greenspace compared to the lowest income group (6–24 million KRW) and the highest income group (≥72 million KRW). Persons with lower incomes already face relatively worse health status, limited access to health care and capital resources, and living in more polluted areas [42,57,58]. The economic crisis caused by COVID-19 could also increase mental health risks. Our results highlight the importance of removing barriers to, and providing opportunities for greenspace, among lower-income communities, which becomes a more crucial issue during the pandemic.

The findings of our study suggest the need for more in-depth discussion on building sustainable guidelines for physical/social distancing in greenspace under a disease outbreak or pandemic rather than prohibiting all access to greenspace. Drawing social-distancing circles in New York’s Domino Park for controlling crowds demonstrates a potential solution that allows urban dwellers to enjoy public greenspace while maintaining physical/social distancing [59]. According to our survey data, when visiting greenspace, 70.2% of the study participants washed their hands frequently, 42.9% used hand sanitizer frequently, 83.5% avoided large groups and gatherings, 60.9% stayed 2 m away from others, 42.5% avoided touching their face, 90.1% wore a face mask, and 49.1% covered their nose and mouth when coughing. Combined with these preventive measures, timely controls on the volume of visitors to greenspace would help diminish the risk of disease spread. Digital tools to monitor the number of visitors in major greenspace can be considered for timely control of crowding in greenspace, while finding the ideal maximum number of visitors per area of greenspace may be challenging. Further research is also needed to investigate the effectiveness of applying such guidelines and safety measures for reducing the potential contributions of visitors to greenspace on disease spread.

Our findings of the decreased visits to greenspace among the study participants in South Korea during the pandemic are inconsistent with the previous reports showing increased mobility to greenspace (e.g., parks) aggregated for some spatial resolutions (e.g., county) in the US and some European cities [30]. The contrasting findings may be due to different measurements of visits to greenspace. We measured changes in visits to greenspace during the pandemic compared to previous years based on perceived changes of individuals. Meanwhile, greenspace near residents as assessed by EVI was not associated with probable cases of MD or GAD. This implies the need to go beyond measuring the amount of greenspace and investigate the role of patterns of using greenspace in relation to mental health benefits.

Our strategy for social media advertisement for our survey was to increase likelihood of participation of the social media users who are exposed to our advertisement campaigns. For this, we had two different campaigns, one using keywords to target Facebook users who set those keywords in their profiles as their interests and the other without keywords set. Our advertisement results show that the number of survey link clicks from Facebook users for the campaign with target words was about 2.5 times higher than the number of link clinks for the campaign without keywords. Therefore, it is likely that our study participants were interested in these topics and may have reported higher frequency of visits to greenspace than general population. An individual’s perception of changes in visits to greenspace may be an important factor that affects the individual’s emotions, moods, and mental health. We controlled for a variable of general frequency and duration of visiting greenspace prior to the pandemic in the statistical model. However, we note that further research with more representative samples of the population will be needed to understand the health effects of greenspace under normal circumstances and a pandemic. Such efforts will aid decision-makers in the prevention of disease spread, urban park management, and urban planning.

One of the limitations of this study is that mental health symptoms during disease outbreaks may not be comprehensively listed in the PHQ-9 or GAD-2, and survey respondents may experience these matters differently from psychiatric perspectives [60]. This may be particularly the case among non-Western cultures. While we intentionally included open-ended questions to elicit life contextual information shown to be critical in past depression research with communities of color [61], we encourage future research to include more locally derived concepts and categories of mental health, along with community-based and partnered approaches. Further, identifying various socio-ecological components of mental well-being (e.g., life satisfaction, security, sense of achievement) of people during disease outbreaks is another important area for further research that can help better connect patients with mental struggles and the culturally responsive mental health care services and interventions. Second, this study is based on a cross-sectional design. Although we adjusted for the history of depression and anxiety in the past year (2019) in the statistical models, we cannot separate whether the link between decreased greenspace and risk of MD and GAD is related to (1) lack of the benefits of greenspace or (2) less incentive and motivation to visit greenspace due to mental well-being, or both. Moreover, the positive RRs of probable MD and GAD in persons with increased visits to greenspace might imply that those persons may be likely to visit greenspace to cope with stress during the pandemic. We assumed that changes in visits to greenspace occurred on a long-term (e.g., months) basis since the initiation of the global pandemic (March 2020) that would trigger accumulated psychological effects to affect the experience of depression and anxiety symptoms began long before the period relevant for survey questions (during the past 2 weeks from the survey period (September–December 2020)). Third, we were not able to distinguish how the effect of social/physical distancing on mental health differed for our study participants, and we did not have data on mobility of each study participant. Additionally, we did not have information on some potential determinants of mental health outcomes such as loss of family or relatives from COVID-19, stressful occupations impacted by the pandemic (e.g., health care workers), and contact with confirmed patients of COVID-19 although we did collect data for the concerns that survey participants reported at a time of survey. Fourth, probable MD and GAD were not based on clinical diagnostic assessments. Nonetheless, the PHQ-9 and GAD-2 have internal consistency and reliability and are commonly used in epidemiologic studies to assess health effects of environmental factors [44]. Fifth, our survey question on the changes in visits to greenspace under the pandemic is vulnerable to a reporting bias that could result in over-reporting of decreases in visits to greenspace due to their concerns for social norms of adhering to the stay-at-home order. Nonetheless, this type of interview bias is considered to be non-differential between the cases and controls of mental health outcomes in this study, which produces a bias towards the null (e.g., conservative risk estimation) [62]. Lastly, we note that living near greenspace and viewing nature from indoors may benefit mental health and well-being but these types of using greenspace were not considered in our survey data.

Our study has several strengths. To our best knowledge, this is the first study to investigate the changes in the way people interact with greenspace under a pandemic and link these changes with mental health. We had data for individual-level risk factors such as education, marital status, history of mental health outcomes, and personal concerns through a survey. Before adding these variables, the crude ORs of decreased visits to greenspace on probable MD and GAD were not statistically significant. Second, our study provided data on type of concerns experienced by survey participants, which are informative to understand potential sources for deteriorated mental health of people under a prolonged pandemic. Third, our study provides potential derivers (e.g., stress relief) and inhibiting factors (e.g., stay-at-home order, fear towards Coronavirus) for visiting greenspace during COVID-19. While human mobility to particular destinations (e.g., work, urban parks, grocery shops) can be tracked by GPS technology and data collection of social media, little is known about how and why people decide to travel to various places. Thus, this epidemiologic study acts as a multidisciplinary work at the intersection of environment, epidemics, mental health, and behavioral sciences.

## 5. Conclusions

Probable cases of MD based on PHQ-9 tool identified from our online survey conducted between September and December 2020 were significantly associated with decreased visits to greenspace among the participants comparing visits during the COVID-9 outbreaks to a pre-pandemic period. We found no significant associations between probable cases of GAD based on GAD-2 and decreased visits to greenspace. The stay-at-home orders and fear and worry for SARS-CoV-2 were associated with decreased visits to greenspace. Job-related and financial concerns were significantly associated with probable MD. Those who used greenspace for social interactions before the outbreak are a high-risk group of deprived mental health benefits of greenspace during the outbreak. While concurrent public health interventions to reduce mental health burdens of COVID-19 are required, greenspace could function as a protective mechanism for social support for mental health when adequate physical distancing is maintained in greenspace. Appropriate policies for using greenspace under the pandemic along with physical/social distancing, along with increased focus on access for lower-income communities, should be considered further by government, communities, and health authorities to improve psychosocial well-being during and independent of a global pandemic.

## Figures and Tables

**Figure 1 ijerph-18-05842-f001:**
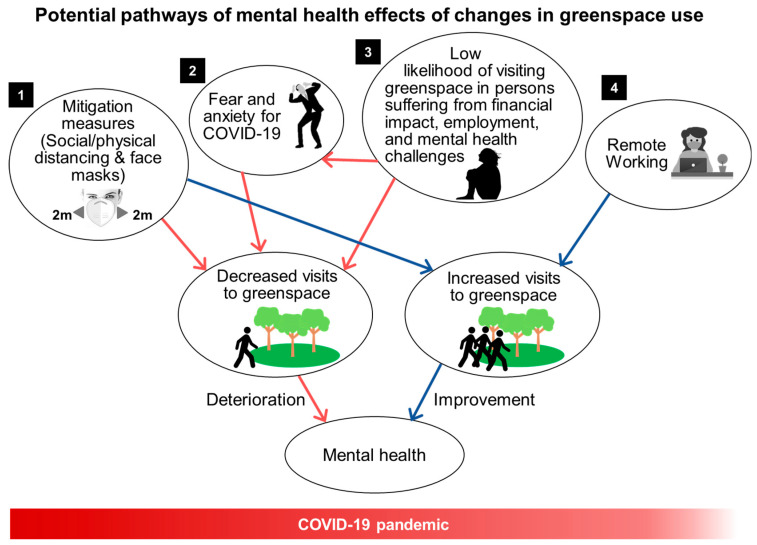
Pathways of changes in interactions with greenspace during the COVID-19 pandemic and their effects on mental health.

**Figure 2 ijerph-18-05842-f002:**
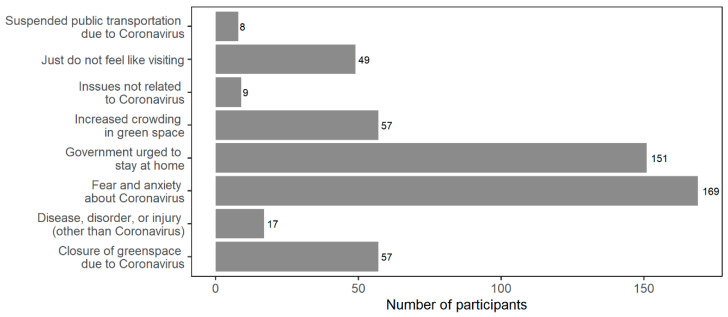
Factors influencing decisions to visit greenspace in 2020 for survey participants with decreased visits to greenspace in 2020 compared to 2019 (*n* = 209, multiple choices were allowed among the answers).

**Figure 3 ijerph-18-05842-f003:**
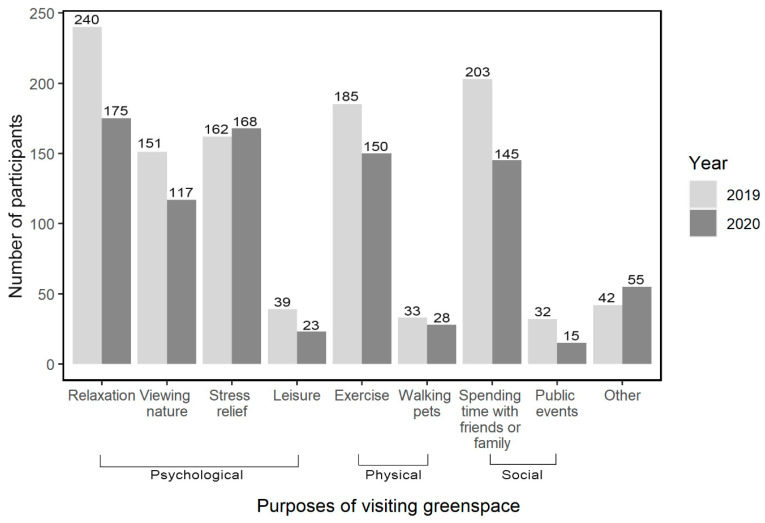
Purposes for visiting greenspace in 2019 (pre-pandemic) and during the COVID-19 pandemic (2020) among the 322 survey participants. Multiple choices were allowed among the answers.

**Table 1 ijerph-18-05842-t001:** Descriptive statistics of survey participants by frequency of probable Major Depression and Generalized Anxiety Disorder (*n* = 322).

Variable	Total	Major Depression			General Anxiety Disorder		*p*-Value
		Yes	No	*p*-Value	Yes	No	
	*n*	*n* (%)	*n* (%)		*n* (%)	*n* (%)	
Total	322 (100)	62 (19.3)	260 (80.7)		48 (14.9)	274 (85.1)	
Gender							
Male	76 (23.8)	8 (2.5)	68 (21.3)	0.014	7 (2.2)	69 (21.5)	0.041
Female	243 (75.9)	52 (16.2)	191 (59.7)		39 (12.2)	204 (63.8)	
Other	1 (0.3)	1 (0.3)	0 (0.0)		1 (0.3)	0 (0.0)	
Age							
19–29 years	115 (35.7)	27 (8.4)	88 (27.3)	0.309	21 (6.5)	94 (29.2)	0.404
30–49	155 (48.1)	27 (8.4)	128 (39.7)		22 (6.8)	133 (41.3)	
50–64	44 (13.7)	8 (2.5)	36 (11.2)		5 (1.6)	39 (12.1)	
≥65	8 (2.5)	0 (0.0)	8 (2.5)		0 (0.0)	8 (2.5)	
Education							
≤Elementary	3 (1.0)	1 (0.3)	2 (0.6)	0.001	0 (0.0)	3 (1.0)	0.036
Middle-High school	81 (26.3)	26 (8.4)	55 (17.9)		19 (6.2)	62 (20.1)	
≥Undergraduate	224 (72.7)	30 (9.7)	194 (63.0)		26 (8.4)	198 (64.3)	
Annual Income							
6–24 million KRW	74 (30.7)	16 (6.6)	58 (24.1)	0.584	11 (4.6)	63 (26.1)	0.137
24–48 million KRW	63 (26.1)	10 (4.1)	53 (22.0)		7 (2.9)	56 (23.2)	
48–72 million KRW	45 (23.2)	11 (4.6)	45 (18.7)		12 (5.0)	44 (18.3)	
≥72 million KRW	48 (19.9)	6 (2.5)	42 (17.4)		3 (1.2)	45 (18.7)	
Marital status							
Married	136 (43.6)	19 (6.1)	117 (37.5)	0.016	14 (4.5)	122 (39.1)	0.048
Single	163 (52.2)	34 (10.9)	129 (41.3)		28 (9.0)	135 (43.2)	
Widowed/divorced/Separated	13 (4.2)	6 (1.9)	7 (2.2)		4 (1.3)	9 (2.9)	
Smoking							
Current smoker	35 (10.9)	9 (2.8)	26 (8.1)	0.590	8 (2.5)	27 (8.4)	0.272
Former smoker	49 (15.2)	9 (2.8)	40 (12.4)		5 (1.5)	44 (13.7)	
Never smoker	238 (73.9)	44 (13.7)	194 (60.2)		35 (10.9)	203 (63.0)	
Alcohol drinking							
Non-drinker	134 (41.6)	28 (8.7)	106 (32.9)	0.367	18 (5.6)	116 (36.0)	0.952
2–4 times per month	71 (22.1)	13 (4.0)	58 (18.0)		12 (3.7)	59 (18.3)	
Once per month	75 (23.3)	11 (3.4)	64 (19.9)		12 (3.7)	63 (19.6)	
2–3 times per week	29 (9.0)	5 (1.6)	24 (7.4)		4 (1.2)	25 (7.8)	
4 times per week	13 (4.0)	5 (1.6)	8 (2.5)		2 (0.6)	11 (3.4)	
Urbanicity							
Urban	176 (57.5)	37 (12.1)	139 (45.4)	0.354	29 (9.5)	147 (48.0)	0.393
Rural	130 (42.5)	21 (6.9)	109 (35.6)		16 (5.2)	114 (37.3)	
EVI *							
<0.13	80 (26.1)	69 (22.5)	11 (3.6)	0.050	67 (21.9)	13 (4.2)	0.064
0.13–0.16	48 (15.7)	40 (13.1)	8 (2.6)		47 (15.4)	1 (3.3)	
0.17–0.19	100 (32.7)	73 (23.9)	27 (8.8)		82 (26.8)	18 (5.9)	
≥0.2	78 (25.5)	68 (22.2)	10 (3.3)		66 (21.6)	12 (3.9)	

Notes. *: Categorization was based on the 25, 50, 75th percentiles. EVI ranges from 0 to 1, with 1 representing more vegetation.

**Table 2 ijerph-18-05842-t002:** Results of multivariable regression analysis: Odds ratio (OR) of major depression and generalized anxiety disorder in relation to changes in visits to greenspace during the pandemic compared to pre-pandemic period, frequency of visits to greenspace in pre-pandemic period, and ZIP-code-level greenness level (EVI).

Variable	Major Depression		Generalized Anxiety Disorder	
	OR	95% CI	OR	95% CI
Changes in visits to greenspace during pandemic compared to pre-pandemic				
Unchanged or increased	1.00		1.00	
Decreased	2.06	(0.91, 4.67) ^†^	1.45	(0.63, 3.34)
Frequency of visits to greenspace in last year (pre-pandemic)				
0–2 times per month	1.00		1.00	
1–2 times per week	0.77	(0.33, 1.78)	0.56	(0.23, 1.37)
3–4 times per week	1.08	(0.34, 3.39)	0.82	(0.24, 2.82)
5 times or more per week	3.08	(0.74, 12.72)	1.29	(0.81, 7.97)
EVI level at residential ZIP code				
<0.17	1.00		1.00	
≥0.17	0.62	(0.29, 1.33)	0.63	(0.28, 1.41)

Notes. ^†^ Significant at a significance level of 0.10.

## Data Availability

The data presented in this study are available on request from the corresponding author. The data are not publicly available due to privacy.

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
