# Peer review of "Impact of Changed Use of Greenspace during COVID-19 Pandemic on Depression and Anxiety"

_ijerph, 2021, doi:10.3390/ijerph18115842_

Round 1

Reviewer 1 Report

Thank you for the opportunity to review this paper focusing on an interesting aspect of areas of impact on mental health as a consequence of the COVID-19 pandemic. On the whole this is a well written paper using sound methodology.

Specific comments

Introduction

  1. Figure 1. Suggest use the word “potential pathways of change” to reflect wording in text

Results section

  1. Suggest highlighting areas of statistical significance in Table 1 in body of article.
  2. Need to check figures for S4 Table and text on page 10 of paper

Line 366 has CI of 1.01, 2.19 which differs from the Table

Line 371 has OR =1.90, 95% CI:1.30,2.77 which differs from the table

Discussion

  1. Suggest starting this section with a synthesis of findings in relations to study aims rather than a presentation on the literature.
  2. What is the relevance of this studies finding to the information presented in the paragraph?
  3. Did the authors consider representing their finding in a diagrammatic way similar to the potential pathways of change presented in the Introduction?

Conclusion

  1. Only reference made to MD outcomes when both MD and GAD were the focus of study. The lack of result for GAD an important finding and consideration of this needed.

Abstract

  1. Finding also need to reflect lack of impact on GAD

Author Response

Reviewer 1

Thank you for the opportunity to review this paper focusing on an interesting aspect of areas of impact on mental health as a consequence of the COVID-19 pandemic. On the whole this is a well written paper using sound methodology.

A: We appreciate your time and suggestions for our manuscript. Following to your suggestions, we revised our manuscript.

Specific comments

Introduction

  1. Figure 1. Suggest use the word “potential pathways of change” to reflect wording in text

A: We changed the text above the graph from “Mental health effects of alternations in interactions with greenspace” to “Potential pathways of mental health effects of changes in greenspace use”.

Results section

  1. Suggest highlighting areas of statistical significance in Table 1 in body of article.
  2. Need to check figures for S4 Table and text on page 10 of paper

A: We added a new sentence of “Women, persons with middle-high school education, and single persons had a significantly higher prevalence of probable MD and GAD.” in lines 302-303. Also, we checked the number in page 10 and Supplementary Table S4 and corrected the errors in Table S4, which responds to the next 2 comments below as well.

Line 366 has CI of 1.01, 2.19 which differs from the Table

A: Thanks for pointing out the typos. We corrected the CIs for BMI associated with probable MD in Supplementary Table S4.  We apologize for this error.

Line 371 has OR =1.90, 95% CI:1.30,2.77 which differs from the table

A: Also, we corrected the CIs for BMI associated with probable GAD in Supplementary Table S4. We apologize for our error.

Discussion

  1. Suggest starting this section with a synthesis of findings in relations to study aims rather than a presentation on the literature.
  2. What is the relevance of this studies finding to the information presented in the paragraph?
  3. Did the authors consider representing their finding in a diagrammatic way similar to the potential pathways of change presented in the Introduction?

A: We shortened some sentences in the first paragraph of the Discussion (explaining previous findings on the mechanisms) and moved them to the Introduction. As a result, the Discussion section was revised to start with our main findings. In particular, we added a new sentence to note that we found the stay-at-home order, physical/social distancing, and fear for COVID-19 to be the major factors for the decreased use of greenspace in our study population, which was suggested by the potential pathways in Introduction. We also revised the paragraphs in the Discussion to describe our findings that are inconsistent with the study aims presented in the Introduction. Please see the changes with track changes in the first paragraph in Introduction and through Discussion. While we do not add a figure for the findings, we discuss them in terms of the pathways described in the Introduction.

Conclusion

  1. Only reference made to MD outcomes when both MD and GAD were the focus of study. The lack of result for GAD an important finding and consideration of this needed.

A: Thanks for the suggestion. Please see that we added a new sentence of “We found no significant associations between probable cases of GAD based on GAD-2 and decreased visits to greenspace” in Conclusion.

Abstract

  1. Finding also need to reflect lack of impact on GAD

A: We added a new sentence of “Decreased visits to greenspace were not significantly associated with GAD (OR=1.45, 95% CI: 0.63, 3.34).” in Abstract.

Reviewer 2 Report

This paper investigates the relation among COVID-19 pandemic, frequency (plus quality and type such as a reason) of access to greenspace and mental health, in South Korea. Based on online survey, the authors find the following facts. (1) COVID-19 pandemic in fact has decreased the use of greenspace in South Korea (2) The index for MD of those who have had less access to greenspace comparing to before pandemic is as about 2 times high as that of those who have accessed to green space at least as frequently as in 2019.

The paper is well-written and provides information of great interest to the readers of this journal. Therefore, below I just list a few minor comments that I hope help improve the paper.

I feel a little bit inconsistency about how to display the results. In fact, in the main text (and figures), the results are sometimes displayed “actual-number-based” (from line 293 to 315, and in Figs 2 and 3) and sometimes “percentage-based” (for instance in lines 320 and 321 that explain Figure3 in which the result is displayed “number-based”). Moreover, in figure 2., x-axis is labeled “number of participants” and in figure 3, y-axis is labeled “Frequency” although they actually show the same content.

In Table2, two types of participants are compared: one is those who have unchanged or increased the access to greenspace and the other is those who have decreased the access. I think, this categorization is a bit unfair because those who unchanged their behavior are categorized as the former group. Why don’t the authors make three groups (increased, unchanged, and decreased) or compare the groups of people purely increased and those decreased?

Discussion is mostly supported by the results, I think. But there is only one concern. The authors write “Our results showed that study participants were more likely to state that they visit greenspace for stress relief after the disease outbreak compared to the previous year…” If this is based on Figure 3, the evidence is too weak since 162 vs 168 does not seem to be a significant difference. It is necessary that a significance test be done as is done for the other results of the paper.

Author Response

Reviewer 2

This paper investigates the relation among COVID-19 pandemic, frequency (plus quality and type such as a reason) of access to greenspace and mental health, in South Korea. Based on online survey, the authors find the following facts. (1) COVID-19 pandemic in fact has decreased the use of greenspace in South Korea (2) The index for MD of those who have had less access to greenspace comparing to before pandemic is as about 2 times high as that of those who have accessed to green space at least as frequently as in 2019.

The paper is well-written and provides information of great interest to the readers of this journal. Therefore, below I just list a few minor comments that I hope help improve the paper.

 A: Thanks for reviewing our manuscript. We have reviewed your comments and revised the manuscript accordingly.

I feel a little bit inconsistency about how to display the results. In fact, in the main text (and figures), the results are sometimes displayed “actual-number-based” (from line 293 to 315, and in Figs 2 and 3) and sometimes “percentage-based” (for instance in lines 320 and 321 that explain Figure3 in which the result is displayed “number-based”). Moreover, in figure 2., x-axis is labeled “number of participants” and in figure 3, y-axis is labeled “Frequency” although they actually show the same content.

A: Thanks for the suggestion. We think that both the actual number and percentage-based findings are informative. To be consistent, we revised the mentioned parts to have both actual number and percentage. These changes can be found in lines 293-301, lines 336-339, and lines 361-362. We changed ‘Frequency’ in Figure 3 to ‘Number of participants’.

In Table2, two types of participants are compared: one is those who have unchanged or increased the access to greenspace and the other is those who have decreased the access. I think, this categorization is a bit unfair because those who unchanged their behavior are categorized as the former group. Why don’t the authors make three groups (increased, unchanged, and decreased) or compare the groups of people purely increased and those decreased?

A: Thanks for the suggestion. We additionally conducted a regression analysis comparing the groups of greenspace use (i.e., unchanged (reference group), increased, and decreased) as suggested. Methods were added in lines 279-280 and the results were added to lines 387-395. We found that OR of probable major depression was significantly higher in persons with decreased visits to greenspace compared to persons with unchanged visits to greenspace. On the other hand, persons with increased visits to greenspace did not show significantly different OR of probable major depression compared to persons with unchanged visits to greenspace. GAD was not associated with either increases or decreases in visits to greenspace. Results of this additional analysis supports the main finding of this study that decreased visits to greenspace was significantly associated with the risk of probable major depression.  

Discussion is mostly supported by the results, I think. But there is only one concern. The authors write “Our results showed that study participants were more likely to state that they visit greenspace for stress relief after the disease outbreak compared to the previous year…” If this is based on Figure 3, the evidence is too weak since 162 vs 168 does not seem to be a significant difference. It is necessary that a significance test be done as is done for the other results of the paper.

A: We agree that our initial wording of this concept is confusing. The reviewer is correct that the fraction of participants who reported visiting greenspace for stress relief in 2019 is not different from the fraction that reported visiting greenspace for stress relief in 2020 (162 participants versus 168 participants. We conducted chi-square test to examine if the frequency of the answers of stress relief differed between 2019 and 2020.  The chi-square was 0.155 and p-value was 0.6935 indicating no significant increases in the answers for stress relief in 2020. However, we wish to highlight that except 'stress relief' and 'other', the frequency of the other purposes all decreased in 2020 compared to 2019, and for some causes decreased significantly. For example, the chi-square test of 'relaxation' or 'spending time with friends or family' showed that the frequency significantly decreased in 2020 (p-value< 0.001). Given the decrease in visits for all other categories (with the exception of “other”), the increase in using greenspace for 'stress relief' during the pandemic is important, even though it is not a statistically significant increase from the previous year.

We modified the original sentence and added new text in the Discussion to clarify this point: “While the increase in visits to greenspace for ‘stress relief’ for the pandemic were slightly increased compared to the previous period, such an increase is notable given that visits for all other specific reasons for visiting greenspace decreased, with statistically significant decreases for some purposes. For example, decreases in using greenspace for 'relaxation' and 'spending time with friends or family' were statistically significant based on chi-square test (p<0.001), whereas visits for 'stress relief' showed a slight increase.” Please see the changes in lines 439-445.